# Computer-aided discovery of novel SmDHODH inhibitors for schistosomiasis therapy: Ligand-based drug design, molecular docking, molecular dynamic simulations, drug-likeness, and ADMET studies

**Saudatu Chinade Ja'afaru**[1,2ʘ]*, **Adamu Uzairu**[1ʘ], **Sharika Hossain**[3], **Mohammad Hamid Ullah**[4], **Muhammed Sani Sallau**[1], **George Iloegbulam Ndukwe**[1], **Muhammad Tukur Ibrahim**[1], **Imren Bayil**[5], **Abu Tayab Moin**[6]*

1 Department of Chemistry, Ahmadu Bello University, Zaria, Nigeria, 2 Department of Chemistry, Aliko Dangote University of Science and Technology, Wudil, Nigeria, 3 Department of Pharmacy, Jahangirnagar University, Savar, Dhaka, Bangladesh, 4 Department of Pharmacy, University of Cyberjaya Medical Science, Cyberjaya Selangor, Malaysia, 5 Department of Bioinformatics and Computational Biology, Gaziantep University, Gaziantep, Turkey, 6 Department of Genetic Engineering and Biotechnology, Faculty of Biological Sciences, University of Chittagong, Chattogram, Bangladesh

ʘ These authors contributed equally to this work.
* sjchinade@yahoo.co.uk (SCJ); tayabmoin786@gmail.com (ATM)

## Abstract

Schistosomiasis, also known as bilharzia or snail fever, is a tropical parasitic disease resulting from flatworms of the Schistosoma genus. This often overlooked disease has significant impacts in affected regions, causing enduring morbidity, hindering child development, reducing productivity, and creating economic burdens. Praziquantel (PZQ) is currently the only treatment option for schistosomiasis. Given the potential rise of drug resistance and the limited treatment choices available, there is a need to develop more effective inhibitors for this neglected tropical disease (NTD). In view of this, quantitative structure-activity relationship studies (QSAR), molecular docking, molecular dynamics simulations, drug-likeness, and ADMET predictions were applied to 31 inhibitors of Schistosoma mansoni Dihydroorotate dehydrogenase (SmDHODH). The designed QSAR model demonstrated robust statistical parameters including an $R^2$ of 0.911, $R^2_{adj}$ of 0.890, $Q^2_{cv}$ of 0.686, $R^2_{pred}$ of 0.807, and $cR^2_p$ of 0.825, confirming its robustness. Compound **26**, identified as the most active derivative, emerged as a lead candidate for new potential inhibitors through ligand-based drug design. Subsequently, 12 novel compounds (**26A**-**26L**) were designed with enhanced inhibition activity and binding affinity. Molecular docking studies revealed strong and stable interactions, including hydrogen bonding and hydrophobic interactions, between the designed compounds and the target receptor. Molecular dynamics simulations over 100 nanoseconds and MM-PBSA free binding energy ($\Delta G_{bind}$) calculations validated the stability of the two best-designed molecules (**26A** and **26L**). Furthermore, drug-likeness and ADMET prediction analyses affirmed the potential of these designed compounds, suggesting their promise as innovative agents for treating schistosomiasis.

**Data Availability Statement:** All relevant data are within the manuscript and its Supporting Information files.

**Funding:** The author(s) received no specific funding for this work.

**Competing interests:** The authors have declared that no competing interests exist.

## Author summary

In an innovative effort to combat schistosomiasis, we have employed a computational drug innovation approach to design a potential treatment options. Schistosomiasis, a parasitic disease affecting millions worldwide, has been a persistent global health challenge. The study, nestled within the broader realm of life sciences, sought to identify a more effective drug using computational methods that analyze highly effective derivatives targeting SmDHODH. This pioneering approach not only accelerates the drug discovery process but also offers a promising avenue for developing targeted treatments. By harnessing computational power, we systematically explored chemical databases to pinpoint compounds with the potential to combat schistosomiasis. The findings hold significant implications for both scientists and non-scientists, as they represent a step forward in addressing a major public health concern. For scientists, this work exemplifies the integration of *in silico* techniques in drug development, while non-scientists can appreciate the tangible impact on improving global health and the well-being of communities affected by schistosomiasis. This research underscores the power of interdisciplinary efforts in advancing our ability to tackle complex health challenges.

## 1. Introduction

Schistosomiasis, a neglected tropical disease (NTD), is transmitted through freshwater snails and is prevalent in sub-tropical Africa, the Middle East, Asia, and Latin America. The disease is endemic in low-income rural communities lacking access to clean water, adequate hygiene, and sufficient healthcare facilities. Sub-Saharan Africa bears the majority of cases, accounting for up to 90%, with an estimated 280,000 annual deaths [1]. The primary species in sub-Saharan Africa are *Schistosoma haematobium*, causing urogenital schistosomiasis, and *S. mansoni*, responsible for intestinal schistosomiasis [2,3]. Schistosomiasis control programs primarily employ community-based preventive chemotherapy, focusing on mass drug administration (MDA) using the only available drug, Praziquantel (PZQ), a broad-spectrum anthelminthic, to reduce morbidity [2]. However, treatment compliance faces challenges due to limited drug options, the potential for the development of drug resistance due to repeated and widespread usage, and PZQ's restricted efficacy against juvenile worms [4–6]. This impacts patient adherence to the medication regimen and increases the risk of reinfection. Therefore, there is a pressing need to develop additional treatment options for schistosomiasis to address its dynamic nature, optimize treatment outcomes, and ensure the long-term success of controlling and eliminating this endemic disease.

Dihydroorotate dehydrogenase (DHODH) is a flavoenzyme responsible for the stereospecific oxidation of (S)-dihydroorotate (DHO) to orotate, constituting the fourth and sole redox step in the de novo pyrimidine nucleotide biosynthetic pathway [7]. Inhibiting the enzyme DHODH in *S. mansoni*, the parasite causing schistosomiasis, offers a promising avenue for therapy. By inhibiting DHODH, the synthesis of pyrimidine nucleotides, essential for DNA and RNA synthesis, is disrupted in the parasite, leading to a depletion of pyrimidine nucleotides crucial for the survival and replication of *S. mansoni* [8]. This deprivation impedes the growth and proliferation of *S. mansoni*, thereby reducing the parasite burden within the host. DHODH inhibitors exhibit selective toxicity towards the parasite while sparing host cells, minimizing potential adverse effects on the host. Additionally, inhibitors of DHODH can potentially synergize with existing antischistosomal drugs, enhancing their efficacy and reducing the

likelihood of drug resistance development [9]. Recent investigations by Renan M. de Mori and colleagues have revealed the structural characteristics of *Schistosoma mansoni* DHODH (SmDHODH) and its human enzyme (HsDHODH), showing notable differences in their conformation [10]. Particularly distinctive in SmDHODH, unlike all other class 2 DHODH structures reported thus far, is the presence of a protuberant domain connecting β6 and βE structural elements [10]. In the realm of drug development, the primary objective is to pinpoint small molecules capable of selectively inhibiting SmDHODH activity in parasites while sparing the human host [11]. Such inhibitors hold the potential to function as antiparasitic drugs, offering a promising avenue for treating infections caused by *Schistosoma mansoni*. Consequently, utilizing the specified characteristics of SmDHODH will enable selective inhibition, presenting an effective strategy for combating schistosomiasis and enhancing the efficacy of current antischistosomal drugs.

Due to the time and cost demands associated with traditional drug design methods, *in silico* drug design has become a widely adopted approach for developing effective treatments [12–14]. Numerous drug design studies now center on Ligand and/or Structure-Based Drug Design [15,16]. In this study, we examined a dataset from ChEMBL and employed ligand based drug design to design derivatives with enhanced activity, high drug scores, and improved binding capabilities to target SmDHODH [17]. This involved the application of various techniques, including Quantitative Structure-Activity Relationship (QSAR), molecular docking, molecular dynamics simulations, drug score computations, and evaluations of pharmacokinetics properties. The development of robust QSAR models enables cost-effective virtual screening of extensive chemical databases, identifying potentially active compounds that meet the criteria for promising drug candidates. The primary aim of this study is to pinpoint and characterize derivatives with the potential to function as inhibitors of SmDHODH, contributing to the control or elimination of schistosomiasis.

## 2. Materials and methods

### 2.1. Materials

The materials used for this research are Toshiba laptop system with processor properties of i5-5200U CPU @ 2.20GHz 2.20 GHz, 8 GB (RAM) on Microsoft Windows 10 Pro Operating System, ChemDraw Ultra 12, Spartan 14V 1.1.2 developed by Wavefunction Inc., PaDel descriptor software, Materials Studio 8.0, Molegro Visual Docker, Discovery Studio Visualizer V. 16.1.0, Osiris property explorer and Desmond program developed by DE Shaw Research. pkCSM and SwissADME online tools were also employed for ADMET and pharmacokinetics predictions of the designed analogs.

### 2.2. Dataset collection, preparation, optimization and activity linearity

A set of thirty-two potential inhibitors for SmDHODH, sourced from the ChEMBL database with ChEMBL ID: CHEMBL4523950, underwent screening to eliminate duplicates and inactive molecules [18–20]. This refinement process resulted in thirty-one compounds selected for further studies. Utilizing the SMILE code provided in the ChEMBL file, 2D structures were generated using ChemDraw software (S1 Table). These 2D structures were then converted into 3D formats employing Spartan 14 software, and their geometric energy was minimized using molecular mechanics force fields (MMFF) [21]. To enhance accuracy, the minimized compounds underwent further geometry optimization through Density Functional Theory (DFT) calculations, specifically utilizing the B3LYP/6-31G* basis set, to achieve a more reliable conformer [22]. The optimized conformers were subsequently saved in *sdf* and *pdb* formats for the determination of molecular descriptors and subsequent molecular docking studies

[16,23]. The inhibitory capacities of the compounds, initially presented as $IC_{50}$ in nanomolar (nM) units, were converted into a logarithmic scale ($pIC_{50}$ = -log $IC_{50} \times 10^{-9}$) to achieve improved data linearity [24]. The 2D structures, biological activities, predicted activities, residuals and respective leverages of the molecules are provided in S1 Table.

## 2.3. Descriptor determination and dataset partitioning

The PaDEL descriptor toolkit was employed to compute essential molecular descriptors influencing the anti-schistosomiasis activities of the derivatives. The 3D structures, saved in *sdf* file format, were imported into the PaDEL software to generate these descriptors [25]. Subsequently, the generated descriptors underwent preprocessing to eliminate highly correlated ones, utilizing version 1.2 of the pretreatment software. After the preprocessing step, the dataset underwent division into modeling and validation sets using the Kennard-Stone algorithm [26]. The modeling set consisted of 22 compounds (70% of the dataset), while the remaining 9 compounds, (30%), were reserved for the external validation test set.

## 2.4. QSAR model construction and validation

A model with the aim of predicting reported experimental data and facilitating the design of new anti-schistosomiasis compounds was constructed using the genetic function approximation approach [27]. This method randomly selects combined descriptors (independent variables) and utilizes biological activities as dependent variables to create models capable of effectively predicting the activities of the dataset. Material Studio software version 8.0 was employed using Multiple-linear regression to formulate the multi-variant equation, and to evaluate the internal validation of the developed model [28]. Afterwards, an external assessment was carried out, and the obtained values were compared and validated against the widely accepted threshold values to ensure the effectiveness and resilience of the constructed model.

**2.4.1. Leverages computation (Applicability domain plot).** The dataset compounds underwent leverage (hi) value calculation (Eq 1) to establish the applicability domain (AD) of the developed model through the utilization of William's plot [29]. This plot offers a graphical representation wherein each compound's leverage value is plotted against its corresponding standardized residual. The diagonal of the hat matrix element denotes the leverage values calculated for both the modeling and validation sets. The standardized residual represents the validated residual estimated from the disparity between predicted and reported experimental activities for both the modeling and validation sets [30]. The threshold for the leverage value is determined through the application of Eq 2.

$$h_i = M_i(M^T M)^{-1} - M_i^T \tag{1}$$

$$h^* = \frac{3(Q+1)}{q} \tag{2}$$

where $h_i$ is the leverage calculation method, $M_i$ is the modeling set matrix of *i*, $M^T$ is the modeling set transpose matrix, M represents the n × k matrix for the validation sets, and $M_i^T$ is the transpose matrix $M_i h^*$. $h^*$ is the warning leverage value, Q is no. of descriptors used to generate the model and q is the modeling set no. of entities.

**2.4.2. Y-scrambling test.** The Y-randomization test is commonly employed to assess the stability of the selected QSAR model by randomly reshuffling the dependent variable (bioactivity) while maintaining the selected descriptors constant [31]. Consequently, the newly generated random models are expected to exhibit low values for the squared regression coefficient ($R_r^2$) and cross-validation coefficient ($Q_r^2$) after multiple iterations, thus validating the

robustness of the original model. Additionally, the coefficient of determination for Y-randomization ($cR^2p$) should exceed 0.5 for a valid model [32].

## 2.5. Ligand-based drug design

Utilizing the selected QSAR model, an *in-silico* screening method was employed to create potential anti-schistosomiasis compounds with improved effectiveness. The lead candidate for drug design was chosen based on the compound exhibiting the highest $pIC_{50}$, low residual value, and favorable pharmacokinetics profile [33]. This selected lead served as the basis for designing new entities, aiming to improve the predicted biological activity and the binding score against the target protein.

## 2.6. Protein preparations and molecular docking studies

The SmDHODH receptor with PDB ID 6UY4 was sourced from the Protein Data Bank (https://www.rcsb.org/). Protein preparations and molecular docking investigations were conducted using Molegro Virtual Docker (MVD) software. The acquired protein was loaded into MVD, where co-crystallized ligands were eliminated, and any identified warnings were rectified [22,23,34]. Following this, a surface was generated, and 5 cavities were identified as potential binding sites. Subsequently, the designed compounds (optimized), were introduced into MVD for the docking study. The selected binding cavity exhibited a volume of 162.816 Å, a surface of 491.241 Å, XYZ coordinates of 17.09; 30.69; 65.73, and a radius of 15 Å. MolDock (Grid) scoring function with a default grid resolution of 0.3 Å, was applied. The docking simulation was independently run 10 times, each with a maximum of 1500 iterations and a population size of 50. Following the completion of the docking procedure, MolDock score, Rerank score, and hydrogen bond energies were generated to assess the ligand-receptor binding strengths. The docked complexes were then saved in PDB format, and their interactions were visualized and interpreted using Discovery Studio software.

## 2.7. Molecular dynamics simulations

Molecular dynamics simulations were conducted on the SmDHODH protein in both its unbound (apo) state and when bound to potential anti-Schistosomiasis agents [35]. The simulations employed the CHARMM36 force field and the Gromacs version 2020 software package [36,37]. To create the simulation environment, the protein-ligand complexes were situated in a rectangular box with a buffer distance of 10 in each direction [38]. The box was then solvated with transferable intermolecular potential with a three-points (TIP3P) water molecules, and $Na^+$ and $Cl^-$ ions were added to mimic a cellular environment [39]. Each system underwent thermal equilibration at a constant temperature of 310 Kelvin through 5000 iterations (equivalent to 10 picoseconds) under the NPT ensemble [40,41]. The Lincs approach was utilized to constrain hydrogen, resulting in a time step of 2 fs [42,43]. Van der Waals forces were investigated using a switching technique with a range of 12–14 and a cutoff value of 14. Long-range electrostatic interactions were computed using the particle mesh Ewald (PME) technique with a maximum grid spacing of 1.2. PME calculations were performed at each iteration without a multiple-time stepping approach, and the barostat's system size changes were set to a target of 1 bar [43]. Numerical integration employed a time interval of 2 femtoseconds. Following the completion of the simulations, output data were analyzed using VMD software, Bio3D, and QTGRACE [44–46].

**2.7.1. Binding free energy calculation using MM-PBSA.** The assessment of binding free energy plays a crucial role in evaluating the stability of ligand-protein complexes [47]. In this study, the MM-PBSA method was employed to compute the binding energy within the

SmDHODH-ligand complexes. This approach takes into account both bonded and non-bonded interactions, including van der Waals and electrostatic forces. The estimation of binding free energy ($\Delta$G) was carried out using Eq (3) through the utilization of the MMPBSA.py script from the AMBER package [48].

$$\Delta G = (G_{complex}) - (G_{protein}) - (G_{ligand}) \tag{3}$$

Where, $G_{complex}$ is the average free energy of the complex; $G_{protein}$ is the average free energy of the receptor in its unbound state; G-ligand is the average free energy of the ligand in its unbound state [49].

## 2.8. Drug score evaluation

The assessment of drug scores involves the incorporation of various factors, including drug-likeness, cLogP, logS, molecular weight, and considerations of toxicity, within a scoring algorithm [50]. This approach aims to provide a quantitative appraisal of the overall potential of the proposed anti-schistosomiasis drug candidates. Osiris Property Explorer was employed to conduct this evaluation [51].

## 2.9. Drug-likeness and ADMET predictions

Following the effective docking of the newly proposed compounds into the binding site of the target receptor, an assessment was conducted to evaluate their suitability as potential drug candidates [52,53]. The designed derivatives underwent scrutiny for drug-like characteristics and ADMET properties. This evaluation was carried out using the pkCSM (https://biosig.lab.uq.edu.au/pkcsm/) and Swiss-ADME (http://www.swissadme.ch/) web tools [54,55].

## 3. Results and discussion

### 3.1. QSAR model construction and validation

The dataset, consisting of 31 derivatives against SmDHODH, was effectively divided into a training set (containing 22 compounds) and a test set (comprising 9 compounds) using the Kennard and Stone algorithm (S1 Table). The training set was utilized to develop a genetic functional algorithm employing the multi-linear regression (MLR) technique as the model equation. The analysis of this genetic functional algorithm explored the physicochemical and structural influences of the compounds under investigation and their corresponding anti-schistosomiasis activities [56]. The proposed Quantitative Structure-Activity Relationship (QSAR) model (shown below) was internally validated, yielding a squared correlation coefficient ($R^2$) of 0.911, an adjusted squared correlation coefficient ($R^2_{adj}$) of 0.890, and a leave-one-out cross-validation squared correlation coefficient ($Q^2cv$) of 0.868 (Table 1). The $R^2$ value of 0.911 indicates that the model captures 91.1% of the variation in the biological activity of the compounds in the training set [57]. The robustness and fitness of the constructed models were also confirmed by an $R^2_{adj}$ of 0.890 as reported in Table 1, and a $Q^2cv$ of 0.868 strongly suggests that the proposed model avoids overfitting [58,59]. The standard error of the model was evaluated to measure its precision in predicting the dependent variable. A lower standard error, like the observed value of 0.402, suggests the model's predictions closely match the actual values. This metric assesses prediction accuracy and reliability, where lower values signify more precise predictions, while higher values imply increased variability and potential limitations in accuracy. Externally, the proposed model underwent cross-validation, yielding a significant predictive squared correlation coefficient ($R^2_{pred}$) of 0.807, meeting the threshold requirements for accepting any proposed QSAR model (Table 1) [58,59]. Importantly, the

**Table 1. The MLR model's statistical parameters in comparison with Golbraikh and Tropsha values.**

| Validation factors | Golbraikh / Tropsha's Threshold | Model values | Validation |
|---|---|---|---|
| Friedman LOF | Low value | 0.724 | Passed |
| R-squared ($R^2$) | > 0.600 | 0.911 | Passed |
| Adjusted R-squared ($R^2_{adj}$) | > 0.600 | 0.890 | Passed |
| Cross validated R-squared ($Q^2cv$) | > 0.600 | 0.868 | Passed |
| Significance-of-regression F-value | A high value | 43.595 | Passed |
| Critical SOR F-value (95%) | $\geq 2.090$ | 3.014 | Passed |
| Standard error | Low value | 0.402 | Passed |
| External R-squared ($R^2_{ext}$) | 0.600 | 0.807 | Passed |
| Average of determination coefficient for Y-randomized data ($R^2_r$) | $< R^2$ | 0.179 | Passed |
| Average of leave one out cross-validated determination coefficient for Y-randomized data ($Q^2_r$) | $< Q^2cv$ | -0.490 | Passed |
| Coefficient for Y-randomization ($cR^2p$) | > 0.500 | 0.825 | Passed |
| Predictive squared correlation coefficient ($R^2_{pred}$) | > 0.600 | 0.807 | Passed |

findings of the proposed QSAR model align strongly with the results of Ibrahim *et al* and numerous other research studies [60–62].

### Proposed QSAR model

$$PIC_{50} = 6.335 * MATS3s + 0.141 * VR2\_Dzp + 9.798 * SpMin3\_Bhm - 8.047 * SpMin4\_Bhs - 2.347$$

The developed model incorporated geometrical and topological descriptors, specifically MATS3s, VR2_Dzp, SpMin3_Bhm, and SpMin4_Bhs. These descriptors played a significant role in providing relevant information and contributions [63], as outlined in S2 Table. Notably, SpMin3_Bhm and SpMin4_Bhs descriptors utilize eigenvalues from the Burden matrix, with SpMin4_Bhs specifically emphasizing the fourth smallest eigenvalue weighted by ionization states. This correlation illustrates the combined influence of atomic masses and ionization states on molecular structure, demonstrating their intertwined roles in defining molecular characteristics and biological activities of the molecules. Pearson correlation statistics were employed for descriptor validation in the proposed model (Table 2). The Pearson correlation analysis conducted revealed values < ±0.9 for all the descriptors which confirms the absence of multicollinearity between any pair of descriptors [50,64]. Additionally, statistical analyses were conducted to assess the model's reliability and robustness. The mean effect (ME) values of each descriptor were determined, representing the average impact of a descriptor on the predicted compound's activities [65]. A positive ME for MATS3s, VR2_Dzp, and SpMin3_Bhm indicated a positive influence on the compound's activity. Thus, adding functional groups that increase the effect of these descriptors would directly enhance the compound's biological activity [66]. Conversely, a negative ME for SpMin4_Bhs suggested a negative influence on the compound's activities. Furthermore, the one-way analysis of variance (ANOVA), was employed to evaluate the significant correlation between anti-

**Table 2. Pearson's correlation coefficient and statistical parameters of developed model.**

| | Pearson's correlation coefficient | | | | Statistical validation parameters | | |
|---|---|---|---|---|---|---|---|
| | *MATS3s* | *VR2_Dzp* | *SpMin3_Bhm* | *SpMin4_Bhs* | *ME* | *P-value* | *Regression coefficients* |
| *MATS3s* | 1 | 0.222 | 0.706 | 0.608 | 0.025 | 0.000 | 6.335 |
| *VR2_Dzp* | 0.222 | 1 | 0.252 | 0.357 | 0.154 | 0.000 | 0.141 |
| *SpMin3_Bhm* | 0.706 | 0.252 | 1 | 0.942 | 1.859 | 8.73E-07 | 9.798 |
| *SpMin4_Bhs* | 0.608 | 0.357 | 0.942 | 1 | -1.039 | 7.02E-08 | -8.047 |

schistosomiasis activities and the descriptors at a 95% confidence level. The reported probability values in Table 2 were all below 0.05 (p < .05) for each descriptor [59]. This indicates the rejection of the null hypothesis, which implies no correlation between anti-schistosomiasis activities and the descriptors in the proposed model [67]. Therefore, the alternative hypothesis suggesting a significant correlation between anti-schistosomiasis activities and the descriptors is accepted. The regression coefficients of each descriptor in the developed model were also examined (Table 2). These coefficients reflect the strength and direction of the relationship between the descriptors and the dependent variable (activity being predicted). Notably, the regression coefficients of MATS3s, VR2_Dzp, and SpMin3_Bhm were positive, implying that an increase in these descriptors is associated with an increase in the predicted anti-schistosomiasis activity. Conversely, the negative regression coefficient of SpMin4_Bhs suggests that a decrease in this descriptor increases the predicted anti-schistosomiasis activity. Notably, the alignment between the mean effect (ME) and regression coefficient adds further confirmation to the reliability of the proposed QSAR model [68].

To reinforce the model's credibility, a Y-Scrambling test was implemented through 50 random trials, involving the random reshuffling of biological activities (dependent) within the training set compounds while keeping the descriptors (independent) unchanged (Fig 1) [31]. The anticipation was that by disrupting the relationship between the descriptors and activity, any correlation observed in the original data would also be disrupted. The Y-scrambling random models yielded an $R^2_r$ value of 0.179, $Q^2_r$ of -0.490, and a $cR^2p$ of 0.825 (Table 1 and Fig 1). The performance metrics from the Y-scrambling test were compared with those of the original model. The original model demonstrates a superior performance compared to the scrambled models. This outcome confirms that the relationships identified in the original model are not a result of random chance correlation [31]. Conversely, the $cR^2p$ value of 0.825 (exceeding 0.5) underscores that the selected model is not a product of chance correlation, further emphasizing its credibility [28].

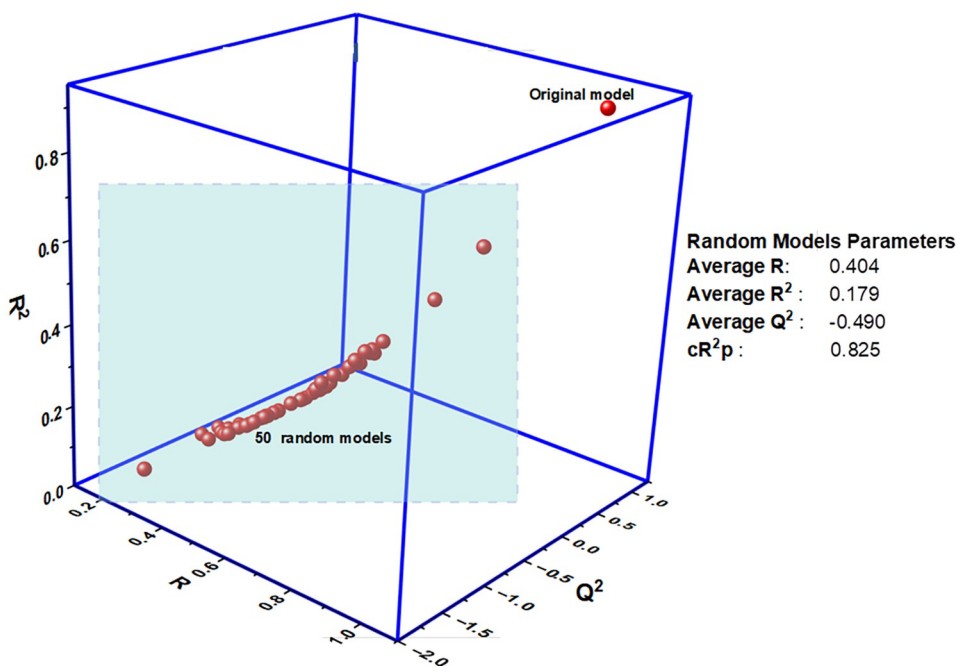

**Fig 1. Y-randomization plot of the develop QSAR model.**

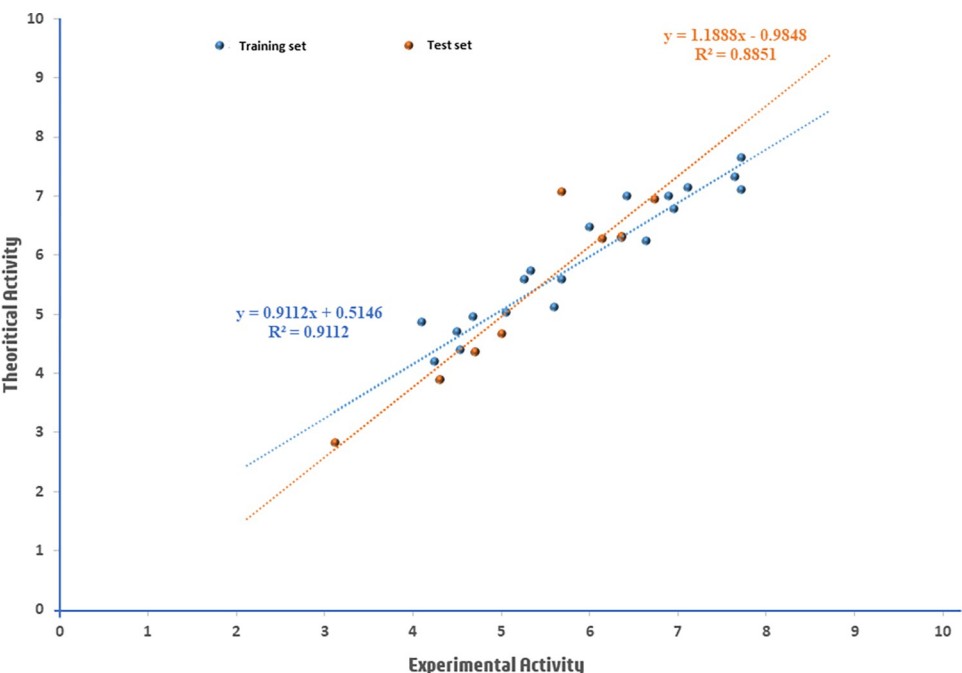

**Fig 2. An activity graph of dataset compounds.**

Fig 2 depicts an activity plot showcasing the predicted $pIC_{50}$ values for both the modeling and validation datasets compared to experimental activity values for inhibiting the SmDHODH enzyme. The primary objective of the activity plot is to distinguish patterns and trends, facilitating an understanding of the structure-activity relationship and aiding in the design of novel compounds with enhanced anti-schistosomiasis activity [5]. In an optimal scenario, a proficient QSAR model would reveal a linear relationship between the predicted and observed biological activities. This indicates the model's ability to precisely capture the structure-activity relationship, ensuring a consistent correlation between the activities [69]. Notably, the plot (Fig 2) demonstrates a strong alignment between the $R^2$ values observed in the developed QSAR and those portrayed in activity plot. The striking similarity between these values and the observed linear relationship, marked by limited scattering and deviations, strongly implies the efficiency of the established model, signifying its robust predictive capacity. Additionally, as reported by Khalifa S. Aminu and colleagues, an $R^2$ value close to 1 underscores the reliability of the selected equation in forecasting the biological activities of novel compounds [34].

Furthermore, an assessment of the model's applicability domain was conducted using Williams's plot, illustrated in Fig 3. The applicability domain (AD) represents the chemical space where a QSAR model is acknowledged as valid and dependable. In this study, all compounds seemed to fall within the specified standardized residual measure of ±3, signifying the absence of outliers [70,71]. However, compounds **1**, **15**, and **20** are identified as influential due to leverage values surpassing the warning threshold of 0.682. These compounds likely possess distinctive structural features that significantly influence the relationships between molecular descriptors and their biological activities [13,71]. Nevertheless, consistent with findings from various studies, an AD plot containing a majority of dataset compounds within the domain is considered valid and reliable [29,72].

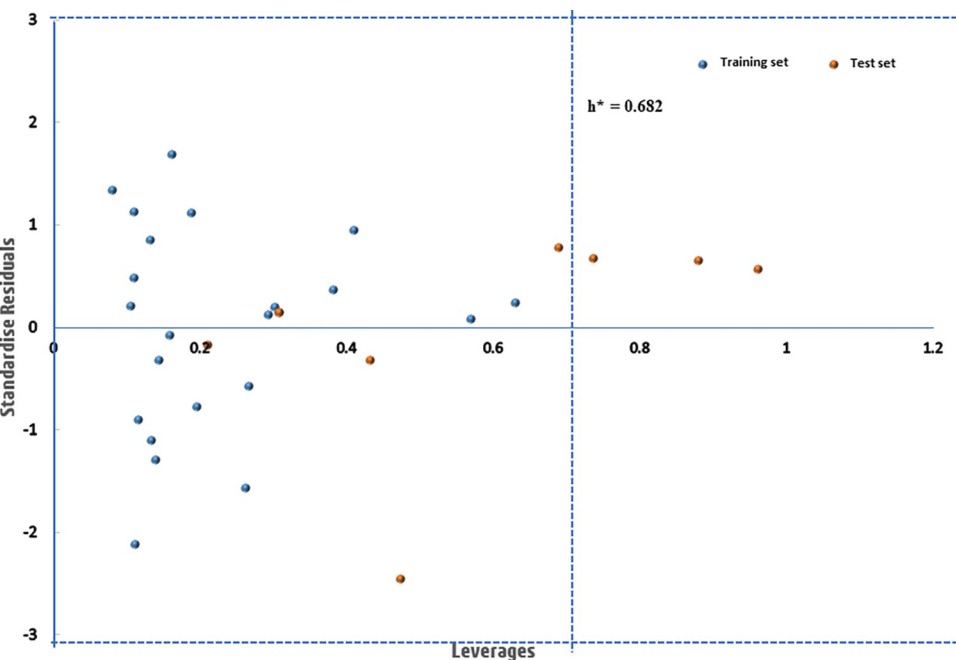

**Fig 3. An applicability domain plots of dataset compounds.**

## 3.2 Ligand-based drug design

Compound **26** is part of the naphthoquinone family, a group of organic compounds characterized by a quinone structure. These compounds are noted for their diverse biological activities, making them promising candidates for various medical applications [73]. Despite reports of toxicity issues linked to this class of compounds [74], several naphthoquinone compounds are already in clinical use, such as the chemotherapeutic anthraquinones as well as mitomycins [75]. Consequently, compound **26** was selected as the principal lead candidate for drug design, with specific positions identified for modifications, as shown in the adopted template (Fig 4). Descriptors such as MATS3s, VR2_Dzp, SpMin3_Bhm, and SpMin4_Bhs were used to guide the selection of substituents for inclusion because of their notable positive and negative mean effect values. Notably, twelve of the newly developed compounds outperformed the lead molecule (**26**) in terms of anti-schistosomiasis activity. This shows that changes based on these characteristics resulted in increased compound activity, potentially presenting these compounds as prospective options for treating schistosomiasis targeting the SmDHODH enzyme.

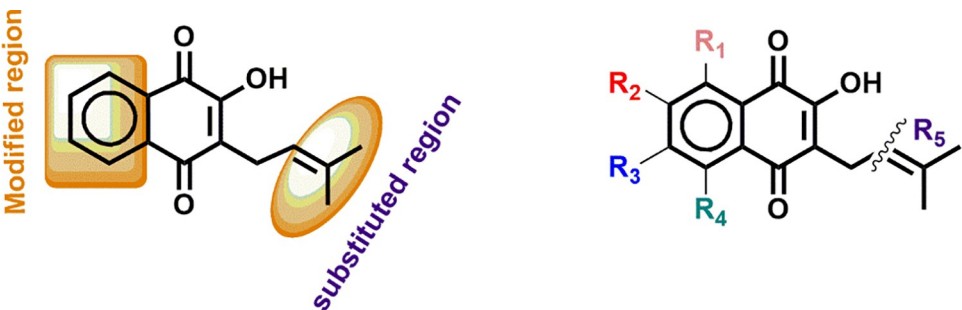

**Fig 4. Structure of the lead compound 26 and the adopted template for ligand-based drug design.**

In the ligand-based design of inhibitors targeting the SmDHODH enzyme, it was observed that incorporating substituents with electron-donating groups (EDG) such as amino ($-NH_2$), methoxy ($-OCH_3$), and hydroxyl (-OH) positively influenced the MATS3s and SpMin3_Bhm molecular descriptors [76]. This positive effect is attributed to the increased electron density contributed by these groups. Conversely, electron-withdrawing groups like nitro ($-NO_2$) and halogens (-Cl, -Br) were noted to potentially decrease electron density, adversely impacting SpMin4_Bhs, and showed promise in enhancing the biological activities of the proposed derivatives [76,77]. The inclusion of these groups resulted in a notable increase in the efficacy of the designed compounds, a phenomenon supported by recent investigations validating the effectiveness of similar substituents [72,78–80]. Structural modifications were carried out on the template structure by substituting the aforementioned groups (-Cl, -Br, $-NH_2$, -OH, $-NO_2$, and $-OCH_3$) at different positions (Fig 4). Notably, the introduction of -Cl, -Br, $-NH_2$, and -OH functional groups at $R_1$ (ortho position) elevated the predicted activities from 7.652 for the lead compound to a range of 7.686–9.149 for the newly designed compounds. Substitutions at positions $R_2$ (meta position) on the aromatic ring had a moderate effect in increasing the biological activities of potential anti-schistosomiasis agents, likely due to the moderate influence of meta-substituents on the resonance structures of the derivative [77]. This effect is evident among the newly designed entities, displaying an activity range of 7.688–7.772 (Table 3). Markedly, compound **26L**, exhibiting the highest activity, featured two methoxy groups substituted at positions $R_2$ and $R_3$. The introduction of $-OCH_3$ groups in meta and para positions on a benzene ring influenced the electron density of the ring through inductive and resonance effects, making it more nucleophilic and potentially impacting reactivity [81]. Overall, all twelve of the newly designed derivatives demonstrated improved inhibitory effects, highlighting the potential of the selected functional groups to enhance the biological activities of the newly designed derivatives.

Moreover, an assessment using drug score was conducted to appraise the potential effectiveness and desirability of a drug candidate. Notably, nearly all of the newly designed compounds exhibited commendable drug score values surpassing both the lead compound (**26**) and the standard drug PZQ, which held a drug score of 0.391 (S1 Fig). The drug scores, falling within the range of 0.12 to 0.77 (Table 3), imply a moderate to relatively high level of efficacy. Side views illustrating the physicochemical characteristics (cLogP, solubility, drug-likeness, and drug score) of the top two designed molecules, **26A** and **26L**, as well as the reference

**Table 3. Newly designed SmDHODH inhibitors with their respective predicted biological activities and drug score.**

| S/N | $R_1$ | $R_2$ | $R_3$ | $R_4$ | $R_5$ | Predicted pIC$_{50}$ | Drug score |
|-----|-------|-------|-------|-------|-------|---------------------|------------|
| 26 | -H | -H | -H | -H | 2-methlpropene | 7.652 | 0.276 |
| 26A | -Cl | -H | -H | -H | 2-methlpropene | 9.149 | 0.625 |
| 26B | -H | -Cl | -H | -H | 2-methlpropene | 7.688 | 0.452 |
| 26C | -Br | -H | -H | -H | 2-methlpropene | 8.642 | 0.431 |
| 26D | -H | -Br | -H | -H | 2-methlpropene | 7.772 | 0.353 |
| 26E | -H | -H | -H | -Br | 2-methlpropene | 7.653 | 0.431 |
| 26F | $-NH_2$ | -H | -H | -H | 2-methlpropene | 8.755 | 0.433 |
| 26G | -OH | -H | -H | -H | 2-methlpropene | 7.686 | 0.718 |
| 26H | -H | -H | $-NO_2$ | -H | 2-methlpropene | 8.401 | 0.402 |
| 26I | -H | -H | -H | $-NO_2$ | 2-methlpropene | 8.020 | 0.403 |
| 26J | -H | -H | $-OCH_3$ | -H | 2-methlpropene | 7.810 | 0.510 |
| 26K | -H | -H | -H | -H | -Cl | 7.810 | 0.126 |
| 26L | -H | $-OCH_3$ | $-OCH_3$ | -H | 2-methlpropene | 10.459 | 0.771 |

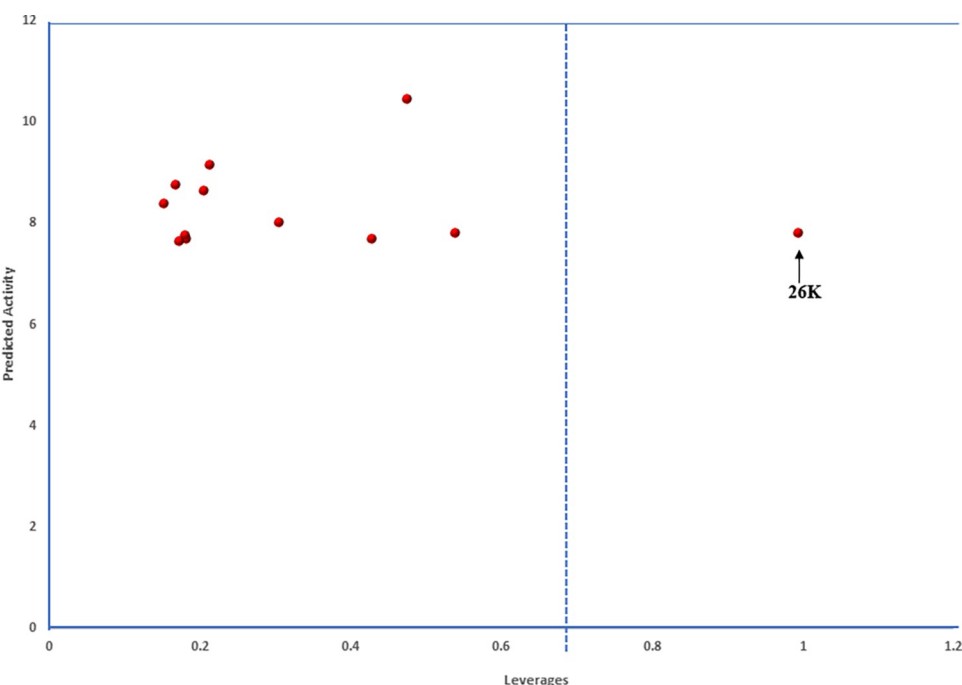

**Fig 5. Leverage plot of newly designed compounds against their predicted activities.**

(PZQ), are depicted in S1, S2, and S3 Figs. The predictions obtained from the OSIRIS Property Explorer are represented and color-coded, with properties posing a significant risk of unintended consequences, such as mutagenicity or poor intestinal absorption, highlighted in red [82]. The green color denotes drug-adherent behavior, while the red color suggests non-adherent conduct. Notably, the figures reveal that the designed compounds exhibited no toxicity risk alerts, displayed in green colors, indicating drug-adherent behavior superior to the standard drug PZQ.

Furthermore, the leverages of the newly designed compounds were computed, utilizing them to construct and analyze the leverage plot presented in Fig 5. This plot serves to prioritize compounds by highlighting the chemical features that have the greatest impact on the desired biological activity, aiding decision-making in the drug development process. Remarkably, eleven out of the twelve designed compounds are situated within the specified AD domain, suggesting their potential as candidates for drug design targeting Schistosomiasis. However, despite observing an increase in biological activity in compound **26K**, its leverage value exceeds the calculated threshold leverage value of 0.682. This discrepancy may indicate that the introduction of substituted chlorine (-Cl) at position-$R_5$ has led to an undesirable effect on the chemical properties of compound **26K**.

### 3.3 Molecular Docking Simulations

The active site of the SmDHODH receptor identified by PDB ID 6UY4 contains key amino acids, including Ser53, Phe92, His50, Ile128, Val358, Arg130, Ala49, Gly46, and Phe357 [10]. These amino acids formed hydrogen bonding and hydrophobic interactions with the SmDHODH inhibitors. Fig 6A illustrates the superimposition docked complex of SmDHODH with the co-crystalized ligand (QLA401), Fig 6B shows a 3D visualization of the superimposed ligand while 6C highlights the key amino acid residues interaction with the QLA401 ligand.

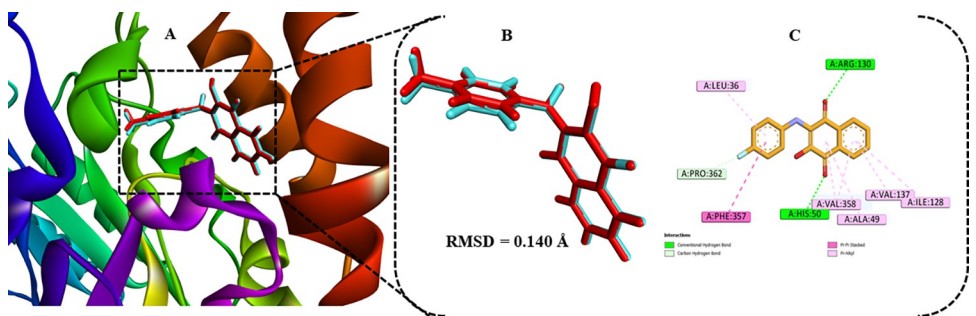

**Fig 6. Superimposed co-crystallized ligand within selected binding cavity of SmDHODH with XYZ coordinated at 17.09;30.69;65.73.** (**A**)-superimposed co-crystalized ligand in complex with 6UY4, (**B**)- Visualization of superimposed co-crystalized ligand, (**C**)- Depiction of active residues interacting with the co-crystalized ligand.

Validating the precision of the docking algorithm is crucial to ensure the accurate binding of ligand molecules to the receptor's active site in a specific conformation. This involves careful selection of the grid box's size and central coordinates. To verify the reliability of the docking approach, the co-crystallized ligand was re-docked, resulting in an RMSD value of 0.140 Å (Fig 6B). This value falls within the accepted standard of an RMSD value below 2.0 Å [23], confirming the accuracy of the docking algorithm. The MVD docking procedure successfully and precisely repositioned the co-crystallized ligand into the SmDHODH binding site, providing evidence for the effectiveness of the docking algorithm.

Docking compound **26** into the SmDHODH optimal binding site revealed notable scores: A MolDock score of -102.332 kcal mol$^{-1}$, a Rerank score of -86.094 kcal mol$^{-1}$, and a hydrogen bond energy of -6.768 kcal mol$^{-1}$ (Table 4). The substantial binding energy emphasizes the strength of the interaction between the ligand and the receptor and a high Rerank score underscores the stability of the formed docked complex. The importance of hydrogen bond energy in establishing overall stability within the ligand-receptor complex is significant [83]. The observed high hydrogen bond energy of -6.768 kcal mol$^{-1}$ indicates a robust interaction between the ligand and the receptor. It is worth noting that Zakari Ya'u Ibrahim and colleagues have previously highlighted that higher values of docking score energies increase the likelihood of the ligand being tightly bound to the receptor's active site [84].

**Table 4. Binding score energies of newly designed SmDHODH inhibitors with their respective RMSD values.**

| S/N | Moldock Score / kcal mol$^{-1}$ | Rerank Score / kcal mol$^{-1}$ | H-Bond / kcal mol$^{-1}$ | RMSD / Å |
|---|---|---|---|---|
| 26 | -102.332 | -86.039 | -6.768 | 0.140 |
| 26A | -107.552 | -91.894 | -9.493 | 1.462 |
| 26B | -103.352 | -87.652 | -6.487 | 0.043 |
| 26C | -102.451 | -84.290 | -6.746 | 1.830 |
| 26D | -108.986 | -90.292 | -6.483 | 1.228 |
| 26E | -110.105 | -95.246 | -5.117 | 6.613 |
| 26F | -104.235 | -93.436 | -3.315 | 0.012 |
| 26G | -104.529 | -94.948 | -5.002 | 0.156 |
| 26H | -111.548 | -48.464 | -11.785 | 1.248 |
| 26I | -101.322 | -77.057 | -8.285 | 0.921 |
| 26J | -111.764 | -93.909 | -9.412 | 0.249 |
| 26K | -93.543 | -80.658 | -10.081 | 0.011 |
| 26L | -113.825 | -82.678 | -8.502 | 0.514 |
| PZQ | -107.604 | -35.633 | -0.587 | 0.004 |

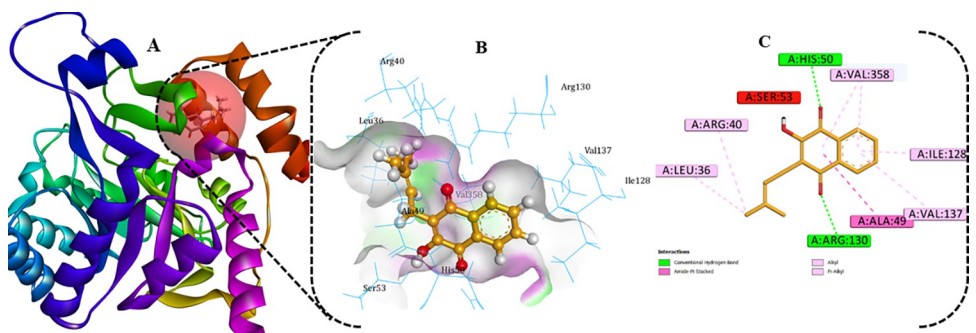

**Fig 7.** (**A**)-SmDHODH-**26** complex full ribbon view; (**B**)-3-D representation and (**C**)-2-D view of the active amino acid residues interactions.

Fig 7 depicts the interactions of the lead candidate (**26**) with the active amino acid residues within the binding site of the target protein. Three conventional hydrogen bonding interactions involving the carbonyl oxygen of the cyclohex-2-ene-1,4-dione scaffold with His50 and Arg130, at distances of 1.902 Å, 2.117 Å, and 1.728 Å, were observered. Furthermore, numerous hydrophobic interactions were identified, specifically with Ala49, Leu36, Arg40, Val358, Ile128, and Val137. It is noteworthy that one unfavorable donor-donor interaction occurred between the ligand's hydroxy moiety and Ser53 possibly due to steric factor. However, nearly all the active amino acid residues of SmDHODH were present within the binding site of the lead compound **26**.

The outcomes of the molecular docking investigations on the ligand-based designed compounds revealed compelling findings. Introducing substituents at the specified positions in the adopted template led to enhanced binding energy scores (Table 4). Notably, the ligand (compound **26L**) with the highest predicted activity at 10.459 also exhibited the top MolDock score of -113.825 kcal mol⁻¹ (Table 4). Compound **26L** stood out as the most effective designed derivative, as it displayed the highest predicted biological activity while maintaining remarkable stability, as indicated by the MolDock score, Rerank score, and hydrogen bond energies. Compound **26L** established numerous interactions with the active amino acid residues within the SmDHODH binding site. Specifically, it participated in four conventional hydrogen bonding interactions involving the carbonyl oxygen of the cyclohex-2-ene-1,4-dione scaffold with His50, Ser50, and Arg130, at distances of 1.876 Å, 2.356 Å, 2.407 Å, and 2.571 Å, respectively. Additionally, seven hydrophobic interactions were observed between 26L and His50, Ala39, Ala49, Val43, Val49, and Val358 (Fig 8A).

The binding interactions of the designed compound (**26A**) with the second-highest predicted activity involved interactions with the target receptor through five conventional hydrogen bonding interactions. These interactions included the carbonyl oxygen of the cyclohex-2-ene-1,4-dione scaffold, hydroxy oxygen, and hydrogen of the hydroxy moiety, engaging with His50, Arg130, and Gly46 at distances of 1.835 Å, 2.126 Å, 2.708 Å, 2.444 Å, and 1.553 Å, respectively. Additionally, a carbon-hydrogen bond interaction occurred between the benzene moiety electron and Ser53 at a distance of 3.048 Å. Other hydrophobic interactions involving Val358, Val137, Leu36, Tyr354, and Ala49 were also observed (Fig 8B). The molecular interactions of the remaining ten designed derivatives are depicted in S4 and S5 Figs.

### 3.4 Molecular dynamics evaluation

To gain a deeper insight into the dynamic behavior and stability of protein-ligand complexes, we examined the results of MD simulations for both the apo form, the lead compound (**26**)-

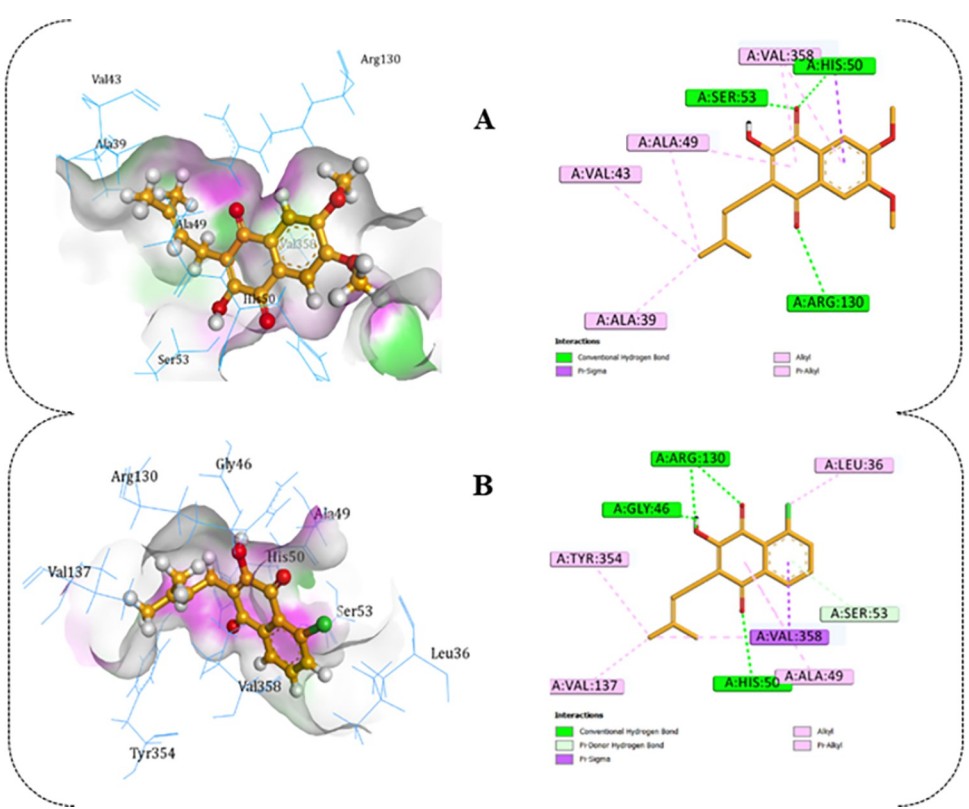

**Fig 8. Molecular docking interactions of the top 2 designed compounds with SmDHODH.** (**A**) SmDHODH-**26L** interactions and (**B**) SmDHODH-**26A** binding interactions.

protein complex and the two best-designed ligand complexes over a 100 ns simulation time [35]. The Root-mean-square deviation (RMSD) serves as a measure for gauging the extent of divergence of a group of atoms from the accurate reference structure of a protein, ligand, or ligand-protein complex. Elevated RMSD values can be indicative of a significant level of instability, stemming from alterations in the conformation of the investigated molecule. For the protein systems **26**, **26A**, and **26L**, the average RMSD values were determined to be 2.759 Å, 2.533 Å, and 2.492 Å, respectively, while the apo protein exhibited an average value of 2.658 Å (Fig 9A). Observably, the RMSD of the apo protein remained relatively constant with minimal fluctuations until approximately 50 to 60 ns, where an increased RMSD was observed. Following this, there was a gradual decrease in the RMSD value, with a noticeable peak at 85 ns, and minimal fluctuations until the end of the simulation period. Noticeably, **26L**-SmDHODH complex system showed a sharp rise within the first 5 nanoseconds, followed by a period of remarkable stability with negligible variations until the 50th nanosecond. At 50 nanoseconds, the RMSD value experienced a 2-angstrom rise but remained constant from that point until the final 5 nanoseconds of the simulation. In contrast, the **26** complex system exhibited a distinct deviation pattern significantly differing from that of **26L**. The RMSD value of the **26** complex system was notably greater than that of both the apo protein and the **26L** complex system. Moreover, the RMSD exhibited a substantial escalation from the start of the simulation, reaching its peak value at 10 ns. Following this period, the RMSD showed a substantial and continuous increase until reaching 60 nanoseconds, contrasting the behavior observed in the **26L** complex system (Fig 9A). Nevertheless, after 60 nanoseconds, particularly during the final 20 nanoseconds of the simulation, the system achieved a state of stability. Additionally, the **26A**

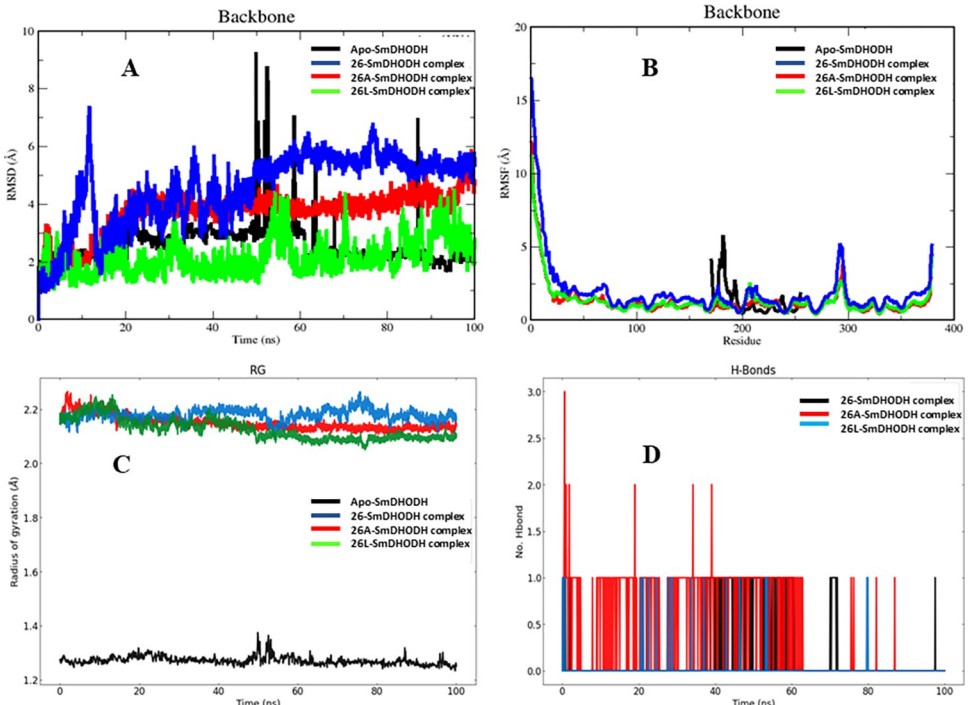

**Fig 9.** Molecular dynamics simulation analysis of studied complexes (A) RMSD plot, (B) RMSF plot, (C) Radius of gyration plot and (D) No. of Hydrogen bond contacts.

complex system displayed the lowest average RMSD value compared to the apo protein and the **26L** and **26** complex systems. Unlike other systems, it maintained a high level of stability within the time range of 20 ns to around 90 ns. In line with the aforementioned observations, it can be concluded that these systems exhibit varying degrees of stability. The **26A** complex system, with its lower RMSD value and stability after 20 ns, and the **26L** system, remaining more stable throughout the simulation, including an RMSD value close to that of the apo protein, support this conclusion (Fig 9A).

Moreover, the root-mean-square deviation (RMSF) values are graphically represented to comprehend the fluctuation at the residue level between the Apo form and ligand complexes (Fig 9B). The RMSF value serves as a metric to discern the rigidity and flexibility of different regions within the protein structure. This method of assessing structural variability in ligand-protein complexes underscores the significance of specific protein residues in these structural changes. Calculating the deviation values for each amino acid position over a 100 ns timescale provides insights into the residues contributing to fluctuations, as illustrated in the RMSF plot presented in Fig 9B [35]. For instance, amino acid positions 170 to 190 in the apo-protein exhibit a deviation of approximately 5 Å, whereas no substantial deviation is observed for the remaining amino acids. This 5 Å difference in the position of amino acids 170–190 could potentially account for the observed variance in (RMSD) of the apo-protein at 50 nanoseconds. Upon comparing the RMSF values of the apo protein with those of the complex systems, it is evident that **26** shows more deviations compared to the other complexes. In the complex systems, amino acid positions 290 to 297 display deviations ranging from approximately 1 Å to 5 Å. The RMSD of the **26** complex system experiences a notable increase after 50 nanoseconds due to positional variations in amino acids. Additionally, examining the ligand-protein RMSF plot for each ligand in complex systems, as depicted in Fig 9B, reveals that the RMSF value at

the C-terminal residues is notably high. This is attributed to the highly reactive and free-moving nature of these tail or end regions of the protein structure.

Additionally, throughout the entire 100 ns simulation period, the parameter known as the radius of gyration (Rg) was employed to assess the compactness of the protein-ligand complexes [35]. An increase in Rg values indicates a decrease in the compactness of the protein structure, signifying heightened flexibility and reduced stability [35]. When comparing the SmDHODH-ligand complex systems, it is evident that the **26L** system displays a smaller radius of gyration in comparison to the 26A and 26-protein complexes (Fig 9C). Interestingly, after 50 ns in the simulation, the Rg value of the **26L** system started to decrease and remained constant until the conclusion of the simulation. In contrast, within the 26A system, the Rg value showed a gradual decline after 15 ns, leading to an augmentation in structural compactness. Consequently, despite both systems demonstrating notably elevated Rg values in comparison to the apo protein, the sustained stability observed throughout the simulation period indicates that the ligand remained securely bound in the active site of the SmDHODH protein.

In addition to examining the RMSD, RMSF, and Rg, we also evaluated the persistence of hydrogen bonds (H-bonds) within protein-ligand complexes throughout the simulation. To comprehend the intermolecular connections, it is essential to conduct a geometric analysis of hydrogen bonding, as these bonds play a critical role in maintaining the structural integrity of biomolecules [35,85]. Moreover, in the context of MD modeling, the formation of hydrogen bonds is pivotal for preserving the stability of complexes. Notably, throughout the entire MD simulation, the number of hydrogen bonds in the ligand-bound states exhibited continual fluctuations, as illustrated in Fig 9D. Specifically, during the molecular dynamics (MD) simulation, the **26A** ligand formed two hydrogen bonds with the SmDHODH protein, while the total number of hydrogen bonds in the **26** and **26L** complexes was one. The graph clearly indicates that the **26A** complex consistently had a higher number of hydrogen bonds throughout the simulation period.

**3.4.1 Principal component analysis (PCA).** Principal Component Analysis (PCA) serves as a valuable method for extracting essential information from Molecular Dynamics (MD) trajectories by discerning global slow motions from local fast motions. In this study, PCA was employed to simulate the significant dynamics of both complex systems and the apoprotein, aiming to explore the nature of interactions among statistically significant conformations discovered along the trajectory [45,85]. The fundamental distinctions within the complexes were elucidated by organizing the primary components into eigenvectors based on their variability. PCA scatter plots of the Apo form, **26**, **26A**, and **26L** systems were generated by projecting simulated trajectories of the protein systems into the two-dimensional subspace spanned by the first three eigenvectors (PC1, PC2, and PC3) (Fig 10). This approach facilitated the investigation of conformational changes in the systems. Fig 10 presents principal component analyses (PCA) revealing that the SmDHODH apo, **26**, **26A**, and **26L** systems contributed 23.09%, 59.37%, 53.35%, and 41.65% of the total variations, respectively. The **26** complex exhibited the highest PC1 value (59.37%), indicating a more substantial number of conformational changes. In contrast, the **26L** complex showed a lower PC1 value (41.65%), suggesting a comparatively smaller alteration in conformation. Moreover, the principal component 1 (PC1) value of the Apo structure (23.09%) is notably lower than that of the **26** complex when compared to the complex systems. This implies that the binding of the **26** ligand results in a less strong interaction, leading to a substantial conformational shift in the Apo form.

**3.4.2 Dynamic cross-correlation matrix (DCCM) analysis.** To investigate the effect of ligand derivatives on the conformational motions of the SmDHODH protein, DCCM analyses were undertaken on all C atoms in the Apo, the **26** complex, **26A**, and the **26L** complex systems using 100 ns simulated trajectories (Fig 11A, 11B, 11C and 11D). The DCCM exhibited a

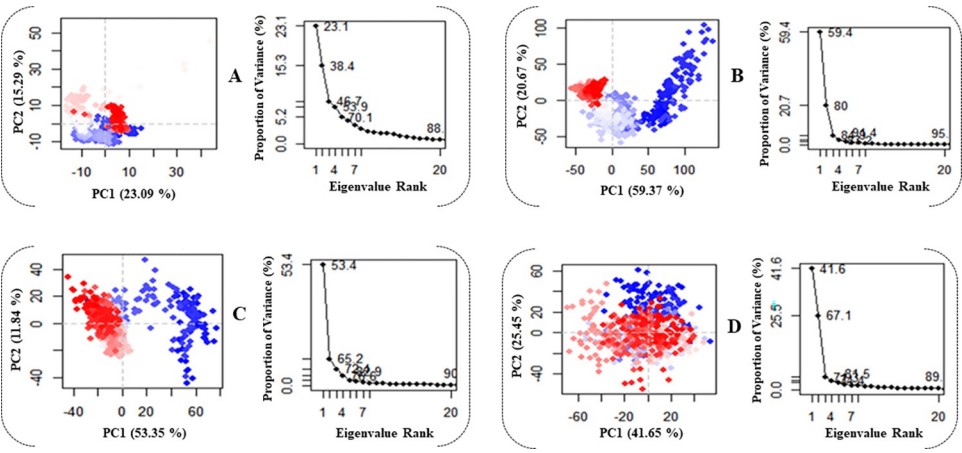

**Fig 10.** Principal component analysis of (**A**) Apo SmDHODH (**B**) SmDHODH-**26** (**C**) SmDHODH-**26A**, and (**D**) SmDHODH-**26L**. Every point corresponds to the protein's conformation on the X and Y axes. The color blue denotes the initial time step, whilst the color white means the middle time step, and the color red indicates the final time step.

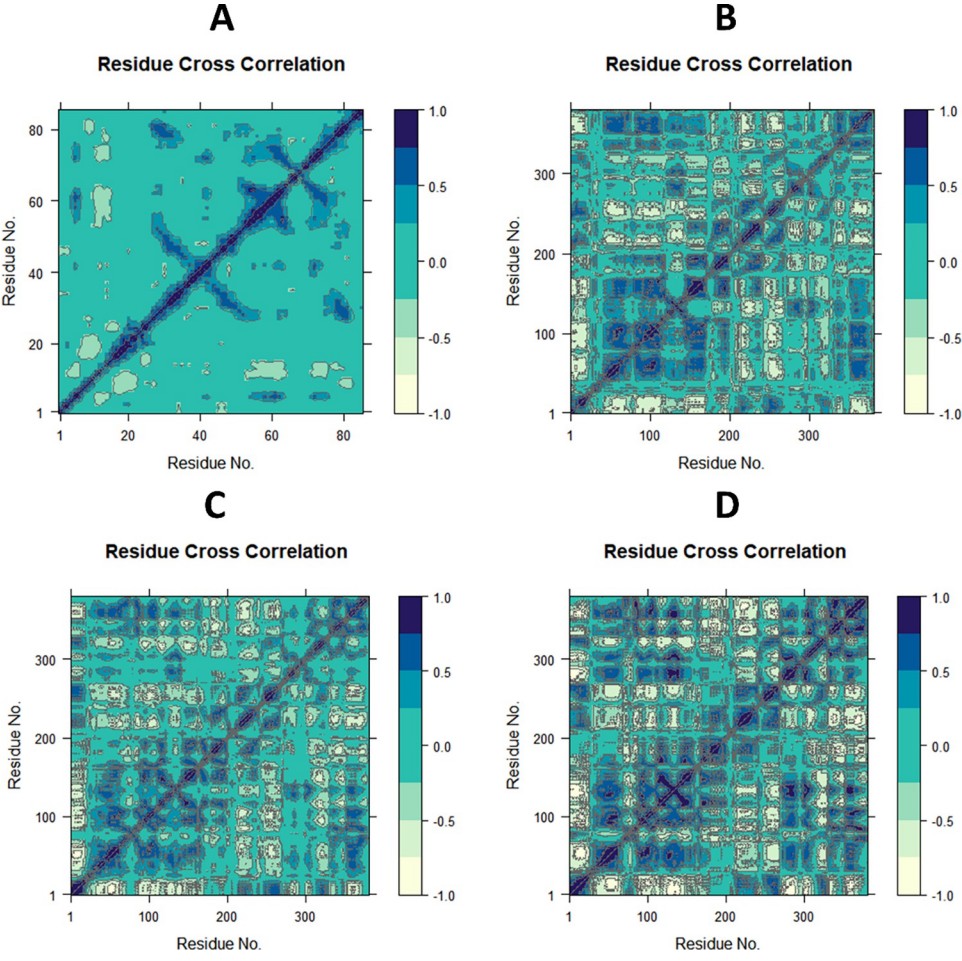

**Fig 11.** Ca-residue cross-correlation matrix for (**A**) Apo SmDHODH, (**B**) SmDHODH-**26** (**C**) SmDHODH-**26A** and (**D**) SmDHODH-**26L** complexes.

comprehensive correlation, encompassing a range of values from − 1.0 to 1.0, with the former indicating a light yellow hue and the latter indicating dark blue hue. It was determined that different shades of color correspond to varying degrees of correlation between residues, with the deeper the color indicating a larger degree of association. The observed correlation coefficient, ranging from − 1 to 1, indicated that residues exhibited either a positive or negative relationship in their movements. A positive correlation indicated that residues moved in the same direction, while a negative correlation indicated that residues moved in opposite directions [35]. After examining the DCCM diagrams of the four systems, it was noted that the coordinated movements displayed by each system were noticeably different. In contrast to the **26A** complex system, in the entire **26L** complex, positively correlated collective movements remained relatively stable, while negatively correlated movements increased significantly. Compared to the **26** complex system, the **26A** complex system experienced a decrease in both positively and negatively correlated movements.

**3.4.3 The binding free energy estimation.** The MM/PBSA approach is a notable method employed for calculating the binding free energy of protein–ligand complexes. Utilizing the MM-PBSA method, the binding free energy of the compounds was determined based on molecular dynamics (MD) trajectories [14,35]. The binding energy ($\Delta G_{bind}$) value was computed, taking into account various protein-ligand interactions, including van der Waals energy ($\Delta EvdW$), electrostatic energy ($\Delta Eele$), and EPB (electrostatic contribution to solvation-free energy via Poisson-Boltzmann) energy (Fig 12). Analysis of the binding free energy for the studied complexes indicated that the **26A**-SmDHODH complex exhibited the highest free energy of -17.37 kJ/mol, compared to **26** and **26L** complexes with energies of -13.92 kJ/mol and -14.85 kJ/mol, respectively. This suggests a robust interaction between the ligand-**26A** and the active site of the target protein, possibly attributed to the increased hydrogen bonding interactions observed in the earlier analysis shown in Fig 10.

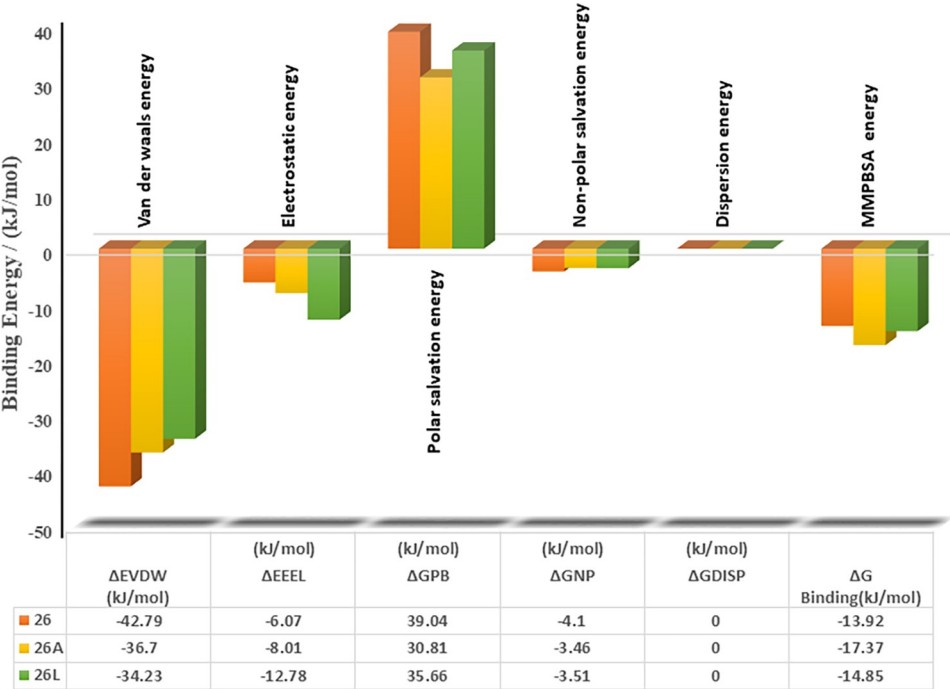

| | ΔEVDW (kJ/mol) | (kJ/mol) ΔEEEL | (kJ/mol) ΔGPB | (kJ/mol) ΔGNP | (kJ/mol) ΔGDISP | ΔG Binding(kJ/mol) |
|---|---|---|---|---|---|---|
| 26 | -42.79 | -6.07 | 39.04 | -4.1 | 0 | -13.92 |
| 26A | -36.7 | -8.01 | 30.81 | -3.46 | 0 | -17.37 |
| 26L | -34.23 | -12.78 | 35.66 | -3.51 | 0 | -14.85 |

**Fig 12. MMPBSA free binding energy plot of studied complexes.**

### 3.5 Drug-likeness and ADMET predictions

The effectiveness of the proposed compounds was assessed through various analyses, including QSAR, molecular docking, and molecular dynamics simulations. These studies demonstrated that the designed compounds exhibited potencies towards inhibiting the target enzyme. Therefore, drug-likeness and ADMET/pharmacokinetic analyses were performed, with the lead compound **26** serving as the reference molecule. To evaluate the likelihood of oral bioavailability and permeability, the designed molecules were scrutinized based on Lipinski's Rule of Five (Table 5) [86]. Adherence to these criteria suggests a higher probability of success as orally active drugs in humans. Notably, all twelve designed compounds met Lipinski's criteria, signifying their potential oral bioavailability. Synthetic accessibility, measured on a scale from 1 to 10, was also examined. Results indicated scores between 2.67 to 3.18 for all designed molecules, falling below the average threshold (Table 5). This suggests that the molecules can be easily synthesized [21]. Additionally, the human absorbance score (HIA), a critical factor for drug effectiveness, exceeded 85% for all the generated compounds, indicating good absorbance levels. Assessment of blood-brain barrier (BBB) permeation revealed that the designed entities demonstrated potential to cross the BBB (Table 5). Moreover, the Boiled-egg plot presented in Fig 13, evaluates the absorption in the gastrointestinal tract and passive diffusion across the BBB. The result predicted showed that all designed compounds fell within the yellow/white regions [87]. This further supports their favorable properties for absorption and penetration across the blood-brain barrier.

Cytochrome P450 (CYP) 450 plays a crucial role in the metabolism of drugs, primarily involving the major liver enzyme system in oxidative metabolism (phase I), as noted by Mustapha Abdullahi and co-workers [28]. Out of the 17 reported CYP families in humans, only four (CYP1, CYP2, CYP3, and CYP4) are associated with drug metabolism. Notably, CYP1A2, CYP2C19, CYP2C9, CYP2D6, and CYP3A4 contribute to the biochemical transformation of over 90% of drugs undergoing phase I oxidative metabolism [88]. Moreover, the majority of

**Table 5. Drug likeness and ADMET parameters of designed compounds.**

| S/N | Drug likeness Parameters | | | | | | ADMET Parameters | | | | | | | | |
| | Lipinski's parameters | | | | | | | | CYP Inhibitors | | | | | | |
| | MW (<500) | MLOGP (<5) | HBD (<5) | HBA (<10) | Lipinski's violation | S/A | HIA | BBB permeant | 1A2 | 2C19 | 2C9 | 2D6 | 3A4 | AMES Toxicity | T-Clearance |
|---|---|---|---|---|---|---|---|---|---|---|---|---|---|---|---|
| **26** | 242.27 | 1.32 | 1 | 3 | No | 2.92 | 95.110 | 0.196 | Yes | Yes | No | No | No | No | 0.175 |
| **26A** | 276.71 | 1.83 | 1 | 3 | No | 2.97 | 93.449 | -0.364 | Yes | Yes | Yes | No | No | No | 0.204 |
| **26B** | 276.71 | 1.83 | 1 | 3 | No | 2.87 | 94.361 | 0.180 | Yes | Yes | Yes | No | No | No | -0.051 |
| **26C** | 321.17 | 1.96 | 1 | 3 | No | 3.07 | 93.382 | -0.366 | Yes | Yes | Yes | No | No | No | 0.112 |
| **26D** | 321.17 | 1.96 | 1 | 3 | No | 3.00 | 94.294 | 0.178 | Yes | Yes | Yes | No | No | No | -0.073 |
| **26E** | 321.17 | 1.96 | 1 | 3 | No | 3.10 | 94.062 | 0.165 | Yes | Yes | Yes | No | No | No | 0.044 |
| **26F** | 257.28 | 0.73 | 2 | 3 | No | 2.98 | 94.082 | -0.050 | Yes | Yes | No | No | No | No | 0.127 |
| **26G** | 258.27 | 0.73 | 2 | 4 | No | 2.98 | 93.593 | -0.038 | No | Yes | No | No | No | No | 0.081 |
| **26H** | 287.27 | 0.3 | 1 | 5 | No | 3.07 | 85.644 | -0.450 | No | Yes | No | No | No | No | 0.081 |
| **26I** | 287.27 | 0.3 | 1 | 5 | No | 3.16 | 85.653 | -0.427 | Yes | Yes | No | No | No | No | 0.016 |
| **26J** | 272.3 | 0.98 | 1 | 4 | No | 3.08 | 95.443 | 0.270 | Yes | Yes | No | No | No | No | 0.196 |
| **26K** | 236.65 | 0.88 | 1 | 3 | No | 2.67 | 94.770 | 0.031 | Yes | No | No | No | No | No | 0.300 |
| **26L** | 302.32 | 0.67 | 1 | 5 | No | 3.17 | 95.607 | -0.319 | No | No | No | No | No | No | 0.206 |

**Key**: MW: molecular weight / $gmol^{-1}$, HBD: hydrogen bond donor, HBA: hydrogen bond acceptor, S/A: synthetic accessibility, T/clearance: total clearance / log ml/min/kg.

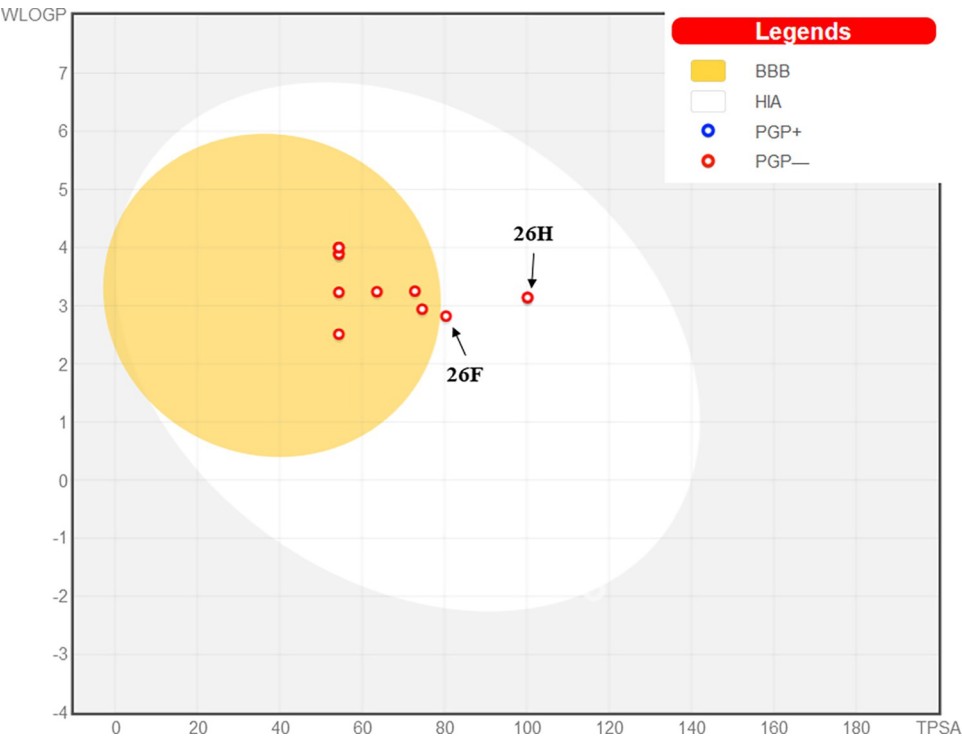

**Fig 13. Boiled egg plot displaying the therapeutic potential of proposed compounds.**

drug metabolism is carried out by two isozymes, namely CYP3A4 and CYP2D6 [89]. The findings indicate that all the designed molecules were anticipated to act as non-inhibitors of CYP2C9 and CYP3A4 which implies that they are unlikely to significantly disrupt the metabolic activity of these enzymes (Table 5). This is crucial for avoiding potential drug interactions and maintaining the normal metabolism of drugs, ensuring their efficacy. Considering the importance of toxicity assessment in drug selection, it is notable that all the designed molecules were predicted to be non-AMES toxic. This underscores a critical aspect of drug development, as non-toxic compounds are generally safer and more suitable for further exploration and potential therapeutic applications. The drug's clearance level indicates how the rate of drug elimination relates to its concentration in the body. The obtained result indicates a low clearance value, suggesting that the proposed compounds remain in the body for a more extended period due to a slower elimination rate. This slower clearance is advantageous, as it implies that the compounds stay in the bloodstream for a prolonged duration, potentially facilitating a more sustained therapeutic effect. In conclusion, these findings suggest that the designed compounds have promising characteristics, making them likely to be effectively absorbed, distributed in the body, and potentially enhancing their therapeutic potential for treatment of schistosomiasis.

## 4. Conclusion

In summary, this *in-silico* investigation introduces twelve novel compounds (**26A**-**26L**) as potential inhibitors of the SmDHODH protein. The estimated pIC$_{50}$ and molecular docking scores (MolDock) for these compounds surpass those of the lead compound and the standard drug Praziquantel. The capability of compounds **26**, **26A** and **26L** to securely bind to the receptor-binding site was confirmed through a 100 ns molecular dynamics simulation.

Furthermore, the designed compounds were predicted to possess drug-like characteristics, meeting Lipinski's rule criteria without exceeding two filtering thresholds, and exhibiting excellent drug scores compared to both the design template and PZQ. Analysis via the leverages plot also affirmed that eleven out of the twelve proposed compounds fall within the specified applicability domain. Consequently, based on these findings, the study recommends further synthesis and experimental validation of these inhibitors as potential SmDHODH inhibitors for Schistosomiasis therapy.

## Supporting information

**S1 Table. Activities and structures of dataset compounds**
(DOCX)

**S2 Table. Interpretation and classes of the molecular descriptors within the developed model**
(DOCX)

**S1 Fig. Profile of the designed compound (26A) showing its potential efficacy.**
(DOCX)

**S2 Fig. Profile of the designed compound (26L) showing its potential efficacy.**
(DOCX)

**S3 Fig. Profile of the Standard drug, Praziquantel (PZQ) showing its efficacy.**
(DOCX)

**S4 Fig.** 3-Dimentional **6UY4** interactions with designed compounds, (**A**) Complex with **26B**; (**B**) Complex with **26C**; (**C**) Complex with **26D**; (**D**) Complex with **26E**; (**E**) Complex with **26F**; (**F**) Complex with **26G**; (**G**) Complex with **26H**; (**H**) Complex with **26I**; (**I**) Complex with **26J**; (**J**) Complex with **26K**.
(DOCX)

**S5 Fig.** 2-Dimentional **6UY4** interactions with designed compounds, (**A**) Complex with **26B**; (**B**) Complex with **26C**; (**C**) Complex with **26D**; (**D**) Complex with **26E**; (**E**) Complex with **26F**; (**F**) Complex with **26G**; (**G**) Complex with **26H**; (**H**) Complex with **26I**; (**I**) Complex with **26J**; (**J**) Complex with **26K**.
(DOCX)

## Acknowledgments

The authors gratefully acknowledge the G.F.S. Harrison Quantum Chemistry Research Group for their support. They also acknowledge the International Foundation for Collaborative Research (IFCR) for providing computational resources, collaborative assistance, research training and fostering international research collaborations, particularly in India, Bangladesh, Philippines and Africa, which were essential for completing this study.

## Author Contributions

**Conceptualization:** Saudatu Chinade Ja'afaru.

**Data curation:** Saudatu Chinade Ja'afaru, Adamu Uzairu.

**Formal analysis:** Saudatu Chinade Ja'afaru, Adamu Uzairu.

**Investigation:** Muhammed Sani Sallau, George Iloegbulam Ndukwe, Muhammad Tukur Ibrahim, Abu Tayab Moin.

**Methodology:** Saudatu Chinade Ja'afaru, Imren Bayil, Abu Tayab Moin.

**Project administration:** Saudatu Chinade Ja'afaru, Adamu Uzairu, Abu Tayab Moin.

**Supervision:** Saudatu Chinade Ja'afaru.

**Validation:** Saudatu Chinade Ja'afaru, Adamu Uzairu, Sharika Hossain, Mohammad Hamid Ullah, Muhammed Sani Sallau, George Iloegbulam Ndukwe, Muhammad Tukur Ibrahim, Imren Bayil, Abu Tayab Moin.

**Visualization:** Saudatu Chinade Ja'afaru, Imren Bayil, Abu Tayab Moin.

**Writing – original draft:** Saudatu Chinade Ja'afaru, Adamu Uzairu, Sharika Hossain, Muhammed Sani Sallau, George Iloegbulam Ndukwe, Muhammad Tukur Ibrahim, Imren Bayil, Abu Tayab Moin.

**Writing – review & editing:** Saudatu Chinade Ja'afaru, Adamu Uzairu, Sharika Hossain, Mohammad Hamid Ullah, Muhammed Sani Sallau, George Iloegbulam Ndukwe, Muhammad Tukur Ibrahim, Imren Bayil, Abu Tayab Moin.

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
