## [Decision Letter · Decision Letter 0]

3 Apr 2024

Dear Mr. Moin,

Thank you very much for submitting your manuscript "Revolutionizing Schistosomiasis Therapy: Targeting Neglected Tropical Disease through SmDHODH Inhibition - A Computational Molecular Modeling Approach." for consideration at PLOS Neglected Tropical Diseases. As with all papers reviewed by the journal, your manuscript was reviewed by members of the editorial board and by several independent reviewers. In light of the reviews (below this email), we would like to invite the resubmission of a significantly-revised version that takes into account the reviewers' comments. 

We cannot make any decision about publication until we have seen the revised manuscript and your response to the reviewers' comments. Your revised manuscript is also likely to be sent to reviewers for further evaluation.

Sincerely,

David Joseph Diemert, M.D.

Academic Editor

jong-Yil Chai

Section Editor

Reviewer's Responses to Questions

**Key Review Criteria Required for Acceptance?**

**Methods**

-Are the objectives of the study clearly articulated with a clear testable hypothesis stated?

-Is the study design appropriate to address the stated objectives?

-Is the population clearly described and appropriate for the hypothesis being tested?

-Is the sample size sufficient to ensure adequate power to address the hypothesis being tested?

-Were correct statistical analysis used to support conclusions?

-Are there concerns about ethical or regulatory requirements being met?

Reviewer #1: Although the authors cited the ChEMBL ID that contains the compounds previously identified as inhibitors of SmDHODH, I believe that the corresponding publications that identified those compounds must be cited:

1 - Nonato, M. C., de Pádua, R. A., David, J. S., Reis, R. A., Tomaleri, G. P., Pereira, H. D. M., & Calil, F. A. (2019). Structural basis for the design of selective inhibitors for Schistosoma mansoni dihydroorotate dehydrogenase. Biochimie, 158, 180-190.

2 - Calil, F. A., David, J. S., Chiappetta, E. R., Fumagalli, F., Mello, R. B., Leite, F. H., ... & Nonato, M. C. (2019). Ligand-based design, synthesis and biochemical evaluation of potent and selective inhibitors of Schistosoma mansoni dihydroorotate dehydrogenase. European journal of medicinal chemistry, 167, 357-366.

Reviewer #2: The study aimed to identify potential inhibitors for the SmDHODH enzyme using computational approaches. Below are comments on the main aspects of the study:

Objectives and Hypothesis:

The study's objectives are reasonably clear, aiming to identify compounds that can act as inhibitors of the SmDHODH enzyme. However, there is confusion regarding the association between the action of these compounds as SmDHODH inhibitors and their subsequent efficacy against schistosomiasis. Clarifying this distinction is necessary for a clearer understanding of the objectives.

Study Design:

The study design, focused on computational approaches, seems appropriate to achieve the stated objectives. The methodology involved selecting compounds previously identified as inhibitors of the enzyme and utilizing them as a basis for computational studies, which included the development of QSAR models for designing potential enzyme ligands. The selected compounds were subjected to molecular docking and molecular dynamics studies to evaluate their interactions with the enzyme. Additionally, the compounds were evaluated for druglikeness parameters and prediction of ADMET properties.

However, it is crucial to recognize that the 31 compounds identified from CHEMBL lack sufficient chemical diversity for a broader study. Additionally, it's noted that the 31 selected compounds were not disclosed, which hinders transparency and reproducibility.

Description of Population:

The target population, in the context of the computational study, consists of 31 compounds identified from CHEMBL. However, these compounds do not possess sufficient chemical diversity for a more comprehensive study. Unfortunately, the specific identities of these compounds were not provided, further complicating the assessment.

Ethical and Regulatory Considerations:

No specific concerns about ethical or regulatory compliance in the context of the computational study were mentioned.

**Results**

-Does the analysis presented match the analysis plan?

-Are the results clearly and completely presented?

-Are the figures (Tables, Images) of sufficient quality for clarity?

Reviewer #1: (No Response)

Reviewer #2: Overall, the results are congruent with the proposed objectives, reflecting a well-executed computational study on identifying potential inhibitors for the SmDHODH enzyme.

**Conclusions**

-Are the conclusions supported by the data presented?

-Are the limitations of analysis clearly described?

-Do the authors discuss how these data can be helpful to advance our understanding of the topic under study?

-Is public health relevance addressed?

Reviewer #1: (No Response)

Reviewer #2: While the computational methodology employed in the study is adequate, the lack of chemical diversity among the identified compounds and the absence of experimental validation are significant limitations. Therefore, any conclusion at this point would be speculative. It is strongly recommended to conduct validation experiments using a more diverse set of compounds to strengthen the study's conclusions and enhance its scientific credibility.

**Editorial and Data Presentation Modifications?**

Reviewer #1: (No Response)

Reviewer #2: (No Response)

**Summary and General Comments**

Reviewer #1: The manuscript is well written and straightforward. Although I would like to see some of these newly identified compounds tested against the enzyme, the results are clear and brings novelty and significance, considering that very little is known regarding inhibition of the S. mansoni DHODH.

Reviewer #2: The manuscript presents a computationally relevant study aimed at identifying potential inhibitors for the SmDHODH enzyme. While the study lacks exploration of artificial intelligence techniques due to data limitations, it utilizes computational approaches effectively. However, the lack of chemical diversity among the identified compounds limits the scope of the study. Despite well-executed computational studies, the absence of experimental validation renders the obtained data speculative, which may not be suitable for publication in a journal like PLOS Neglected Tropical Diseases.

One notable concern is the conflation between SmDHODH inhibition and its consequent anti-parasitic action, which also requires validation. Although the computational studies were well-conducted, they should be complemented by experimental validation to strengthen the credibility of the findings.

Overall, the manuscript may overpromise compared to what it delivers. The title and author summary may exaggerate the significance of studying SmDHODH inhibition. Additionally, while the authors mention the problem of schistosomiasis in Africa, it's important to note that schistosomiasis is a global issue, and a broader perspective on the disease should be provided.

Furthermore, there is currently no clear evidence of resistance development to praziquantel, and the SmDHODH enzyme is not yet a validated target. However, identifying inhibitors for SmDHODH is an important step towards achieving this objective and warrants further investigation.

In conclusion, while the manuscript presents valuable computational insights, it falls short in terms of experimental validation and broader contextualization of the research. Addressing these shortcomings would significantly enhance the manuscript's scientific rigor and relevance.

PLOS authors have the option to publish the peer review history of their article (what does this mean?). If published, this will include your full peer review and any attached files.

Reviewer #1: No

Reviewer #2: No
---

## [Decision Letter · Decision Letter 1]

5 Jul 2024

Dear Dr. Moin,

Thank you very much for submitting your manuscript "Computer-aided discovery of novel SmDHODH inhibitors for Schistosomiasis therapy: Ligand-based drug design, Molecular docking, Molecular dynamic simulations, drug-likeness, and ADMET studies" for consideration at PLOS Neglected Tropical Diseases. As with all papers reviewed by the journal, your manuscript was reviewed by members of the editorial board and by several independent reviewers. In light of the reviews (below this email), we would like to invite the resubmission of a significantly-revised version that takes into account the reviewers' comments. 

We cannot make any decision about publication until we have seen the revised manuscript and your response to the reviewers' comments. Your revised manuscript is also likely to be sent to reviewers for further evaluation.

Sincerely,

David Joseph Diemert, M.D.

Academic Editor

Jong-Yil Chai

Section Editor

Reviewer's Responses to Questions

**Key Review Criteria Required for Acceptance?**

**Methods**

-Are the objectives of the study clearly articulated with a clear testable hypothesis stated?

-Is the study design appropriate to address the stated objectives?

-Is the population clearly described and appropriate for the hypothesis being tested?

-Is the sample size sufficient to ensure adequate power to address the hypothesis being tested?

-Were correct statistical analysis used to support conclusions?

-Are there concerns about ethical or regulatory requirements being met?

Reviewer #1: (No Response)

Reviewer #3: Please see the Summary and General Comments session.

**Results**

-Does the analysis presented match the analysis plan?

-Are the results clearly and completely presented?

-Are the figures (Tables, Images) of sufficient quality for clarity?

Reviewer #1: (No Response)

Reviewer #3: Please see the Summary and General Comments session.

**Conclusions**

-Are the conclusions supported by the data presented?

-Are the limitations of analysis clearly described?

-Do the authors discuss how these data can be helpful to advance our understanding of the topic under study?

-Is public health relevance addressed?

Reviewer #1: (No Response)

Reviewer #3: Accept after major revisions.

**Editorial and Data Presentation Modifications?**

Reviewer #1: (No Response)

Reviewer #3: Please see the Summary and General Comments session. Accept after major revisions.

**Summary and General Comments**

Reviewer #1: (No Response)

Reviewer #3: The article presents the findings of an in silico drug design, which intelligently integrates various complementary molecular modeling techniques. This integration facilitates the rational and accelerated development of drugs, as demonstrated by the results presented. However, the article could benefit from minor adjustments, particularly by carefully reviewing the text for repetitive passages. These repetitions may occur either in terms of expression (such as the consistent use of identical terms throughout the text, e.g., 'leverage') or in terms of redundant information.

Additionally, some crucial points require clarification:

1. The authors mention the existence of resistance to praziquantel in the abstract. However, while this possibility has been observed in laboratory settings and there are instances of low responsiveness to treatment in African countries, to the best of my knowledge, this resistance is not conclusively proven. If the authors possess new information on this matter and can provide references, it would be important to disseminate it to the readership.

2. Statistical data lack accompanying errors (e.g., values of q2 and q2adj). Including this information is crucial.

3. The authors introduce the constructed QSAR model and its validation, elucidating how independent variables influence pIC50. Nevertheless, no explanation has been given regarding the chemical and/or physical significance of the variables associated with the model's action. Therefore, it is imperative to explain the chemical and physical significance of parameters such as MATS3s, VR2_Dzp, SpMin3_Bhm, and SpMin4_Bhs.

4. Moreover, it appears to me that some of the parameters listed as important are indeed correlated (e.g.: MATS3s and SpMin3_Bhm; SpMin3_Bhm and SpMin4_Bhs). Please, explain those correlations coeficients.

5. Both the 31 compounds used as the basis for constructing and validating the QSAR model and the 12 newly proposed analogs lack their chemical structures in the provided information. It is crucial for these structures to be included in the main text rather than relegated to supplementary materials (especially the newly proposed ones).

6. In this context, Figure 4 suggests that these compounds are naphthoquinones. If indeed true, these compounds might carry potential toxicity. Solely relying on in silico-based toxicity predictions in this section of the article warrants reconsideration in light of existing literature information on these compounds.

Suggested Article’s DOI:

a. 10.3390/molecules190914902

b. 10.1080/01480545.2022.2104306

c. 10.1080/01480545.2022.2104306

d. 10.1080/01480545.2022.2104306

Finally, I'd like to make one last comment about the study: a 100 ns molecular dynamics simulation might be too brief to adequately demonstrate the stability of the analyzed complex. While I acknowledge that the authors may not have access to equipment for longer simulations, if feasible, extending the simulation to at least 300 ns would be advantageous.

PLOS authors have the option to publish the peer review history of their article (what does this mean?). If published, this will include your full peer review and any attached files.

Reviewer #1: No

Reviewer #3: No
---

## [Decision Letter · Decision Letter 2]

13 Aug 2024

Dear Dr. Moin,

We are pleased to inform you that your manuscript 'Computer-aided discovery of novel SmDHODH inhibitors for Schistosomiasis therapy: Ligand-based drug design, Molecular docking, Molecular dynamic simulations, drug-likeness, and ADMET studies' has been provisionally accepted for publication in PLOS Neglected Tropical Diseases.

Best regards,

David Joseph Diemert, M.D.

Academic Editor

Jong-Yil Chai

Section Editor

Reviewer's Responses to Questions

**Key Review Criteria Required for Acceptance?**

**Methods**

-Are the objectives of the study clearly articulated with a clear testable hypothesis stated?

-Is the study design appropriate to address the stated objectives?

-Is the population clearly described and appropriate for the hypothesis being tested?

-Is the sample size sufficient to ensure adequate power to address the hypothesis being tested?

-Were correct statistical analysis used to support conclusions?

-Are there concerns about ethical or regulatory requirements being met?

Reviewer #3: (No Response)

**Results**

-Does the analysis presented match the analysis plan?

-Are the results clearly and completely presented?

-Are the figures (Tables, Images) of sufficient quality for clarity?

Reviewer #3: (No Response)

**Conclusions**

-Are the conclusions supported by the data presented?

-Are the limitations of analysis clearly described?

-Do the authors discuss how these data can be helpful to advance our understanding of the topic under study?

-Is public health relevance addressed?

Reviewer #3: (No Response)

**Editorial and Data Presentation Modifications?**

Reviewer #3: (No Response)

**Summary and General Comments**

Reviewer #3: Authors have fulfilled almost all the requirements or suggested alterations in the manuscript which improved a lot, so it should be considered to the publication at PLOS Neglected Tropical Diseases.

PLOS authors have the option to publish the peer review history of their article (what does this mean?). If published, this will include your full peer review and any attached files.

Reviewer #3: No

---

## [Editor Report · Acceptance letter]

28 Aug 2024

Dear Dr. Moin,

We are delighted to inform you that your manuscript, "Computer-aided discovery of novel SmDHODH inhibitors for Schistosomiasis therapy: Ligand-based drug design, Molecular docking, Molecular dynamic simulations, drug-likeness, and ADMET studies," has been formally accepted for publication in PLOS Neglected Tropical Diseases.

Best regards,

Shaden Kamhawi

co-Editor-in-Chief

Paul Brindley

co-Editor-in-Chief
